# Treatment of HMG-CoA Lyase Deficiency—Longitudinal Data on Clinical and Nutritional Management of 10 Australian Cases

**DOI:** 10.3390/nu15030531

**Published:** 2023-01-19

**Authors:** Susan Thompson, Ashley Hertzog, Arthavan Selvanathan, Kiera Batten, Katherine Lewis, Janelle Nisbet, Ashleigh Mitchell, Troy Dalkeith, Kate Billmore, Francesca Moore, Adviye Ayper Tolun, Beena Devanapalli, Drago Bratkovic, Cathie Hilditch, Yusof Rahman, Michel Tchan, Kaustuv Bhattacharya

**Affiliations:** 1Sydney Children’s Hospitals’ Network, Westmead, NSW 2145, Australia; 2Faculty of Medicine and Health, Westmead Campus, University of Sydney, Westmead, NSW 2154, Australia; 3Faculty of Medicine and Health, University of New South Wales, Randwick, NSW 2031, Australia; 4Mater Hospital, Brisbane, QLD 4101, Australia; 5Westmead Hospital, Westmead, NSW 2145, Australia; 6PathWest Laboratory Medicine, Nedlands, WA 6009, Australia; 7Women and Children’s Hospital, Adelaide, SA 5006, Australia

**Keywords:** HMG Co A Lyase, fat oxidation, leucine oxidation, low-protein diet, low-fat diet, ketones, 3 Hydroxybutyrate, hyperammonemia, metabolic decompensation, liver failure

## Abstract

3-Hydroxy-3-Methylglutaryl-CoA Lyase (HMGCL) deficiency can be a very severe disorder that typically presents with acute metabolic decompensation with features of hypoketotic hypoglycemia, hyperammonemia, and metabolic acidosis. A retrospective chart and literature review of Australian patients over their lifespan, incorporating acute and long-term dietary management, was performed. Data from 10 patients contributed to this study. The index case of this disorder was lost to follow-up, but there is 100% survival in the remainder of the cases despite several having experienced life-threatening episodes. In the acute setting, five of nine patients have used 900 mg/kg/day of sodium D,L 3-hydroxybutyrate in combination with intravenous dextrose-containing fluids (delivering glucose above estimated basal utilization requirements). All patients have been on long-term protein restriction, and those diagnosed more recently have had additional fat restriction. Most patients take L-carnitine. Three children and none of the adults take nocturnal uncooked cornstarch. Of the cohort, there were two patients that presented atypically—one with fulminant liver failure and the other with isolated developmental delay. Dietary management in patients with HMGCL deficiency is well tolerated, and rapid institution of acute supportive metabolic treatment is imperative to optimizing survival and improve outcomes in this disorder.

## 1. Introduction

3-Hydroxy-3-Methylglutaryl-CoA Lyase (HMGCL) is a mitochondrial enzyme that catalyzes the cleavage of HMG-CoA to acetoacetate and acetyl-CoA, the final common step of ketogenesis and leucine catabolism [1] (Figure 1). HMGCL deficiency was first described in 1976, and the gene was identified and cloned in 1993 [2,3,4]. A recent systematic review identified 211 cases, with an overall mortality of 16% [5]. A recent report from Saudi Arabia showed that all patients in their cohort of 62 patients had neurological complications [6]. Approximately 40% of reported patients presented in the newborn period with a rapidly progressive encephalopathy [5]. Many experienced recurrent episodes of decompensation and had impaired development. Later-onset presentations have also been described, including acute decompensation and fatality during labor or intercurrent illness [7]. There have been a variety of treatment reports, but none over the complete lifespan.

### 1.1. History

The first case reported in the literature was presented in Western Australia in 1976 with typical findings of hypoglycemia and acidosis [2]. Follow-up on this case (at age 4 ½ and 10 years) indicates he followed a carbohydrate-based diet with a moderate protein restriction [8,9,10]; the few episodes of metabolic decompensation were managed at home with additional glucose [8,9]. Further potential cases were subsequently reported from Australia [11]. Several other cases were reported worldwide in the 1980s and 1990s [12,13,14]. Larger series have since been published, the condition being particularly common in Saudi Arabia, Portugal, and South America [1,12,15,16,17]. Several case reports indicate initial treatment has included L-carnitine supplementation and dietary restriction of leucine or protein [18]. Some reports include maltodextrin supplementation, leucine-free amino acid mixture, and/or fat restriction [8,10]. 

Case reports from New South Wales for patients with HMGCL deficiency utilized adjunctive sodium D,L 3-hydroxybutyrate (S-DL-3OHB) (Veriton Pharma, Weybride, UK) to manage decompensation. This practice has extended to other centers in Australia [19,20]. 

### 1.2. Review of Treatment

Grünert and Sass recently compiled a detailed case review of clinical presentation and management. They reported 7.7% of 117 patients as having no specific dietary treatment [5,7]. Of the remainder that stated dietary prescription, 43.8% reported intake of low-leucine formula or low-protein diets, and 54.3% followed a low-leucine/protein and low-fat diet. Only 1% had a fat-restricted diet without protein restriction. Some patients also consumed additional carbohydrates in the form of maltodextrin or cornstarch. In total, 78% of 109 patients reported L-carnitine supplementation. In their earlier review of 35 patients, 66% received either a leucine restricted diet (60–100 mg/kg/day) or a protein restriction [5]. Leucine-free medical formulas were introduced to 54%, and median protein intake ranged from 1.7 g/kg/day at three months of age to 0.8 g/kg/day at 16 years. In 31% of patients, fat intake was restricted to 20–30% energy intake. Tube feeding was utilized in 18% of the group, and 83% were supplemented with maltodextrin during infections. Data had been ascertained as a cross-sectional data set, so it is difficult to know if individuals transitioned from one style of management to another over time. Dietary macronutrient distribution is not clear from many reports [5,12,21]. 

### 1.3. This Study

This review of Australian treatment aims to provide detailed description of dietary strategies to provide guidance on the management of this severe condition both in acute circumstances and long-term.

## 2. Materials and Methods

Retrospective chart review of patients from five centers in Australia identified by 31 March 2022. Informed consent was taken per local jurisdictional regulations. Local human research ethics approval was obtained.

## 3. Results

Eleven patients were identified, and one family withdrew consent. Four patients have been published previously [11,19,20,22]. Two of these cases (Patients 1 and 2) were included in the recent international series by Grünert and Sass in 2020 [5], with the other two (Patients 4 and 5) being published in prior abstracts [19]; studies on Patient 4 were also published in 2020 (Patient 4) [20]. Individual case reports are available as online Appendix A. Presenting features are indicated in Table 1. The index case of this disorder (Patient 1) published in 1976 has been lost to follow-up in rural Australia, having provided monitoring samples until the age of 35 years. Per published reports, he had been managed with a high-carbohydrate, moderate-protein diet until last published at age 10 years [9,10]. No further clinical update is available. The age range of the remainder of our cohort is 2 months to 39 years, with follow-up on the oldest patient (Patient 2) from 3 months of age to 39 years. Across the remainder of the cohort, there is 100% survival.

Patient 2, currently aged 39 years, had five acute metabolic admissions until 16 months of age and has been metabolically stable since. From diagnosis at four months of age, dietary protein was restricted to 1 g/kg/day using maltodextrin. At five months of age, 120 mg/kg/day of leucine was prescribed, with the remaining protein in the diet from a branch-chain amino acid free medical formula. At two years, uncooked cornstarch was tried but not accepted. She was reported as also having Usher Syndrome, complicating intellectual outcome reporting [22]. For the three adults on whom we have data (Patients 1–3), despite presenting in early childhood, they have not had an acute presentation for the last 11–22 years.

### 3.1. The Impact of Newborn Screening

Newborn screening in Australia has potentially identified HMGCL deficiency since 1998. Six patients (Patients 5–10) were born after this time, with two presenting clinically, prior to availability of a positive newborn screening result (Patients 5 and 6). The screening test is typically collected on day two and reported by day eight. Patients 8 and 10 had clinical symptoms including weight loss and lethargy when recalled after the newborn screening result but had not clinically presented to the hospital at the time. 

Patient 9 presented with severe liver dysfunction at nine months. Patient 7 was identified aged 2.5 years with developmental delay but no apparent metabolic decompensation. Neither Patient 7 nor Patient 9 was identified by newborn screening, even when data were analyzed retrospectively with knowledge of the diagnosis. All four patients that were identified by newborn screening have subsequently presented with acute metabolic decompensation, including hypoglycemia and acidosis, indicating that prospective diagnosis has allowed pre-emptive treatment.

### 3.2. Treatment

Treatment and outcome are presented in Table 2. None of the cases utilize a leucine-free medical formula except patient 2 who was prescribed a branched chain free medical formula in infancy. The children have been managed with a moderate-protein, low-fat (and carbohydrate supplemented) diet, which has been liberalized as they have grown older. All of the patients have had maltodextrin-based emergency management plans. Four patients have used high-dose S-DL-3OHB (900 mg/kg/day) as part of their acute management plan. There have been no pregnancies managed yet in this cohort. Five patients are prescribed L-carnitine at 100 mg/kg/day, with another two patients on lower doses. Three children take nocturnal uncooked cornstarch. None of the adults manage nocturnal fasting with uncooked cornstarch or any other therapy.

## 4. Discussion

This study demonstrates the detailed components of treatment, which are summarized in the above results and the online case reports. HMGCL deficiency is a severe disorder associated with acute metabolic decompensation and long-term neurological sequelae. Rapid resuscitation when unwell may be lifesaving and avert complications. Long-term dietary management with a low-protein diet has been widely reported to have prevented decompensation. Some centers also restrict fat intake.

### 4.1. Data on Illness and Fasting

Stable isotope studies were reported in HMGCL-affected non-identical twins in a control setting, while fasting, and during an unwell period [18,23]. The latter demonstrated increased protein turnover when unwell, more so than during fasting or when well. Concomitant urinary metabolite excretion showed equivalent gross increase in 3-HMG excretion when fasting and in illness, compared to control conditions. However, there was greater turnover of leucine and excretion of 3-methylgutaconate (3-MGC) and 3-hydroxyisovalerate (3-HIVA) in illness compared to fasting. The authors also conclude that 3-HMG must arise from both fat and leucine oxidation. 

Extrapolation from this isolated study would suggest that patients are more likely to tolerate fasting than illness, which seems to be supported by the clinical data of patients being stable for long periods of time. However, sudden acute collapse is reported, potentially precipitated or compounded by physiological stress [5,19,24]. Patient 6 had a hypoglycemic seizure the morning after omitting her dinner and nocturnal dose of uncooked cornstarch and ketones. She responded promptly to dextrose infusion and frequent S-DL-3OHB dosing and was discharged home the next day. Patient 4 had life-threatening decompensation with hyperammonemia and cerebellar tonsillar herniation into the foramen magnum, having not been managed (nor seen by) medical services for 14 years [19,20]. This was precipitated by intercurrent illness. Fasting-related hypoglycemia can be promptly managed but illness- (and pregnancy-) related issues are potentially much more severe and protracted [20,24,25]. This reinforces the need for careful management of intercurrent illness.

### 4.2. Clinical Presentation in This Cohort

The majority of patients in this cohort presented with typical acute acidosis and encephalopathy, but two presented atypically. Patient 7 presented with developmental delay without apparent acute decompensations. This has been reported as an atypical presentation of HMGCL deficiency [26,27]. Patient 9 presented with fulminant liver failure with a peak AST of 21,992 U/L (0–97), ALT of 18,213 U/L (0–58), and INR of 9.8 (1.0–1.2). Profound encephalopathy could not be attributed to hyperammonemia as the contemporary ammonia in serum was 22 µmol/L (0–50). Urine organic acids were indicative of HMGCL deficiency; glucose delivery was immediately doubled and 900 mg/kg/day S-DL-3OHB administered, leading to rapid resolution of clinical and biochemical findings. Whilst liver dysfunction is reported, liver failure is atypical for this disorder, and this type of presentation is more in keeping with a fatty acid oxidation disorder [6,28]. This case was the only one in our cohort that had quantifiable, though reduced, ketones in the urine and blood. The hepatology team considered the presentation typical of paracetamol intoxication, but paracetamol levels were not suggestive of this. No viral etiology was ascertained.

### 4.3. Pathophysiology

Like many inborn errors of metabolism, the pathophysiology of HMGCL deficiency can be considered in terms of potentially toxic metabolites (leucine) accumulating and deficiency of product (ketone bodies). Counterregulatory compensation due to hypoglycemia is hugely impaired because both leucine catabolism and fat oxidation are affected, leading to secondary metabolic dysfunction [29,30,31]. In situations of energy deficiency, this has proven to be catastrophic in many instances. In the leucine oxidation pathway, metabolites such as 3-MGL and 3-HIVA may be notably elevated. 3-HIVA and 3-HMG have also been observed in patients with MRI spectroscopy, indicating that these proximal metabolites (Figure 1) could be involved in pathophysiology [32]. Intramitochondrial accumulation of acetyl-coA can also deplete Coenzyme A recycling for other processes [33]. The relation, in terms of symptomatology, between 3-MGC accumulation due to leucine oxidation and as a marker of mitochondrial dysfunction remains unclear.

Of the disorders of leucine catabolism, severe isovaleric acidemia has the same predilection for rapid acute encephalopathy with hypoglycemia, acidosis, and hyperammonemia. Whilst there are some similar cases of 3-methylcrotonyl-CoA carboxylase (3MCCC) and 3-methylglutaconyl-CoA hydratase deficient patients, there are equally many asymptomatic individuals with these biochemical conditions. As a consequence, the role of 3-HMG, 3-MGL, and 3-HIVA in the pathophysiology of these disorders must be better defined [8,31,34,35]. Traditionally, the hyperammonemia in organic acidemias has been thought to arise from inhibition of N-acetylglutamate synthase but may also arise from insufficiency of Krebs cycle intermediates, favoring α-ketoglutarate production from glutamate [36,37,38]. The Reye Syndrome phenotype is more typically associated with fat oxidation disorders, and this could arise from ketone body synthetic insufficiency in the context of hypoglycemia [39].

### 4.4. Utilization of Ketone Bodies

Plasma, cerebrospinal fluid, and magnetic resonance spectroscopic studies have demonstrated clear appearance and likely utilization of B-OHB in two children with diabetes after administration of S-DL-3OHB [40]. Dramatic changes in cardiac and CNS function have been seen when this has been utilized with multiple acyl-CoA dehydrogenase deficiency [41,42]. As such, our center in Sydney—with hospital drug committee approval and federal government notification—has been using ketone salts as rescue therapy for acute metabolic decompensation in disorders of ketone body synthesis since 2010 [19,20]. This has been used more broadly in Australia since that time.

#### Stereoisomer Utilization

S-DL-3OHB in this study has been administered as the racemic mixture. Animal data suggests that cytoplasmic L-3OHB is more prevalent in myelin synthesis [43]. Therefore, the abnormal white matter appearance that is widely reported could be associated with this deficiency in the untreated state [15,44]. The appearance does not necessarily correlate with either progressive disease or cognitive function in our patients, although this has not been systematically studied. Using stable isotope studies from human neonates and infants, production of D-3-hydroxybutyrate appears rapidly in under four hours of fasting [45]. This study suggests that neonates produce and utilize ketones equivalent to approximately 25% of basal energy requirements (10 kcal/kg/day), which supports clinical observations that presentation in the neonate is particularly common. The study used the labelled d-isomer as the basal infusion, and utilization was extrapolated from this, indicating metabolism of this stereoisomer. Taken together, these data suggest that both isomers may have different important physiological effects, and further studies of this in HMGCL deficiency would be informative.

### 4.5. Development of Practice Guidelines

Our practice guidelines have been developed based on our opinion and experience (Table 3) and the published evidence discussed above.

#### 4.5.1. Glucose and Complex Carbohydrates

The mainstay of management of acute decompensations has been glucose administered in the home environment as maltodextrin or as intravenous dextrose in the hospital. This has been administered at rates typically above calculated endogenous glucose production rates (i.e., 6–9 mg/kg/min in infants, 5 mg/kg/min in children and 2.5 mg/kg/min in adults) [46]. Whilst hypoglycemia is a feature of this disorder, it is a late manifestation, and ketone body response may be absent or blunted at this time, leading to acute tissue damage. We have promoted the use of low-glycemic-index foods during the day and uncooked cornstarch at doses between one to two grams per kilogram at night as an additional security measure, especially in young children (over 12 months of age). This would delay fasting glycogenolysis and fat oxidation and consequent metabolic decompensation. Patient 6 had a hypoglycemic seizure the morning after a busy day at age three. She had inadvertently missed her dinner, night-time starch, and ketone dose.

#### 4.5.2. Fat- and Protein-Restricted Diet

When first identified, this condition was considered a leucine catabolic disorder and dietetic strategies aimed at reducing leucine specifically, with several case reports indicating benefit with leucine or protein reduction [9,11,21,47]. However, it is clear that ketone body insufficiency occurs subsequent to failure to utilize acetyl-coA from fatty acid oxidation as well. In practical terms, our team has restricted both fat and protein through the addition of maltodextrin to the newborn feeding regimen. As children grow older, complex (low-glycemic-index) carbohydrates are introduced and families are taught a low-fat, moderate-protein diet. These patients are at risk of micronutrient deficiencies, especially of essential fatty acids and fat-soluble vitamins and potentially of calcium, iron, and B12. It is difficult to be certain of the efficacy of this dietary management, but the addition of maltodextrin and micronutrient supplementation is the simplest way that we have found to mitigate risk. Regular surveillance of micronutrients should occur throughout childhood and adult life. Several older children have adhered less stringently to this diet plan and have more liberal intake; we continue to recommend macronutrient distribution at estimated energy requirements, thereby not recommending excessive intake of fat or protein.

#### 4.5.3. Ketones

Sodium D,L 3-hydroxybutyrate has been used in this cohort, but other mixed ketone salts, and nonracemic and ketone esters could potentially be used. Patient 4 had 4 hourly ketones administered from six weeks of age to two years. There may have been improvement in brain MRI initially, but this was not sustained. We did not demonstrate efficacy of long-term use of ketone salts in this one patient. He currently is 13 years old and has normal cognition and neurology.

In contrast, the acute use of high-dose S-DL-3OHB appeared to preserve cognition after Patient 3 had developed cerebellar tonsillar herniation. In Patient 4, acute use of ketones with intravenous dextrose was associated with no metabolic decompensations in childhood after protracted oral herpes simplex infection. In Patient 8, there appeared to be rapid restoration of cognitive function within 12 h of treatment and subsequent improvement from fulminant liver failure. Use of ketones in the acute setting is biologically plausible in this disorder and seems to have benefitted these three patients. High-dose ketones are used both in resuscitation and pre-emptively in the sick day protocol to prevent decompensation.

#### 4.5.4. L-Carnitine

There is no published consensus on optimal carnitine dose and this variation is reflected in the Australian cohort. Similar to other inborn errors of metabolism, such as glutaric aciduria or methylmalonic aciduria, 100 mg/kg/day of L-carnitne is typically used in childhood and is decreased, in terms of dose/kg, in older individuals [48,49].

## 5. Conclusions

Though rare, HMGCL deficiency can be a severe life-threatening disorder. Lifelong vigilance and specialist care are required for this disorder, as there are a number of fatalities that have been reported at different ages. There are several patients in Australia across a large age range. Thus, practice guidelines have been developed by our metabolic multidisciplinary teams. It is of paramount importance to manage acute decompensation promptly—we use glucose-based infusion and ketones. More study is required on the utility of macronutrient restriction and ketone supplementation, correlating their effect with the brain MRI and developmental outcomes. Studies should also address the impact of physiological stress, such as pregnancy or exercise, on these patients.

## Figures and Tables

**Figure 1 nutrients-15-00531-f001:**
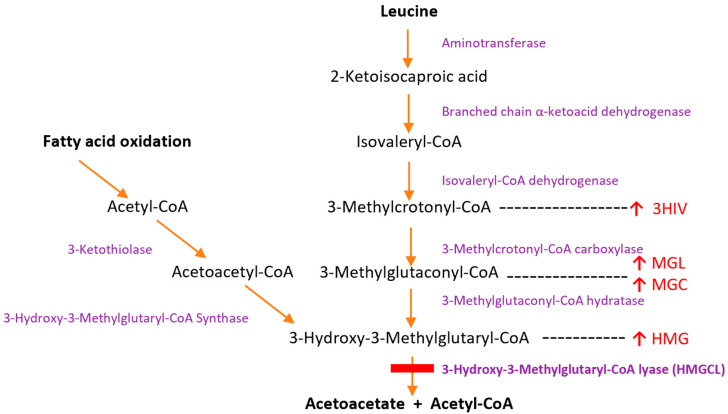
Graphical illustration of the various enzymes and metabolites involved in leucine catabolism and ketone body synthesis. HMG-CoA Lyase is the enzyme involved in the final step of mitochondrial ketone body production from both fatty acid oxidation and leucine catabolism. Reduced or absent function of this enzyme causes accumulation of HMG (3-hydroxy-3-methylglutyrate), MGC (3-methylglutaconate), MGL (3-methylglutarate), and 3HIV (3-hydroxyisovalerate) in bodily fluids.

**Table 1 nutrients-15-00531-t001:** Initial presentation of Australian patients with HMG-CoA Lyase deficiency.

ID	Age (Years)	Age at Diagnosis	Presentation	Ethnicity	Positive NBS	Initial Urinary Organic Acid Elevations						Initial Plasma Acylcarnitine Elevations	
						3HMG	3MGC	3MGL	3HIVA	DCA	3OHB	C5OH (µmol/L)	C6DCA (µmol/L)
1	NA	7 months	Hypoglycaemia, acidosis	Caucasian	NA	+ + +	+ + +	+ + +	+ + +			Not done	Not done
2	39	3 months	Hypoglycaemia, acidosis	Chilean	NA	++	+ + +	+ + +	+ + +	-	-	Not done	Not done
3	33	4 days	Hypoglycaemia, acidosis	Caucasian	NA	++	+ + +	+	++	-	-	Not done	Not done
4	27	10 months	Hypoglycaemia, acidosis	Slovakian	NA	+ + +	+ + +	+	+	-	-	Not done	Not done
5	13	3 days	Acidosis, hyperammonaemia Hypoglycaemia	Israeli	Y	+ + +	+ + +	+ + +	+ + +	-	-	2.2 (<0.2)	1.8
6#	6.5	4 days	Weight loss > 10%	Iraqi	Y	++	+ + +	+	-	-	-	0.36 (0.01–0.15)	0.18 (0.03–0.11)
7	3	2.5 y	Developmental delay	Lebanese	N	+	+ + +	+	-	-	-	0.02 (0–0.1)	0.12 (0–0.08)
8	2	6 days	Weight loss > 10%, vomiting	Pakistan	N	+ + +	+ + +	++	+				
9	1.8	9 months	Liver failure	Caucasian	N	++	+ + +	+	++	+ + +	+ + +	0 (0–0.1) *	0.03 (0–0.08) *
10	0.8	D7 (NBS)	Mild vomiting	Lebanese	Y	+ + +	+ + +	++	+	-	-	1.26 (0–0.1)	1.2 (0–0.08)

Diagnosis and demographic features of patients with HMG co A Lyase deficiency. NBS—newborn screening; 3HMG—3-Hyrdoxy-3-methylglutrate; 3MGC—3-methylglutaconate; 3MGL—3-methylglutarate; DCA—dicarboxylic aciduria; 3OHB—3-hydroxybutyrate. C5OH—Hydroxyisovalerylcarnitine; C6DCA—3-methylglutraylcarnitine. + slight elevation, ++ moderate elevation, +++ gross elevation, - not present, # NBS taken day 2; baby presented day 4, and diagnosis confirmed from NBS on day 9 when transferred to tertiary center. * Total and free carnitine were very low when testing performed, so all acylcarnitines were low.

**Table 2 nutrients-15-00531-t002:** Treatment and outcome of patients with HMG-CoA Lyase deficiency.

ID	Macronutrient/Diet Rx on Diagnosis	Current Age	Current Protein Intake	Current Fat Intake (% Total Energy)	Ketones/Day when Well	Unwell Ketones	Night Rx	Current Carnitine Supplement	Admissions since Diagnosis	Intellectual Outcomes #
1	Leucine restriction	NA	-	-	-	-	-	-	-	-
2	Leucine restriction	39	0.5 g/kg/day	18%	None	None	None	None	5	Moderate delay
3	-	33	0.8 g/kg/day	-	None	None	None	100 mg/kg/day to start	9	Normal
4	Fat and protein restriction	27	1 g/kg/day	~10%	None	None	None	1.5 g per day	1	Normal (blind) *
5	Fat and protein restriction	13	1.4 g/kg/day	8%	Nocte	4 hrly	None	100 mg/kg/day	5	Normal
6	Fat and protein restriction	7	3.4 g/kg/day	8–20%	Nocte	4 hrly	UCCS 1.5 g/kg/day	100 mg/kg/day	2	Normal
7	None	3.5	Normal	Normal	Nocte	4 hrly	None	100 mg/kg/day	0	Moderate delay
8	None	2.1	Normal	Normal	Twice daily	4 hrly	UCCS 1 g/kg/day	40 mg/kg/day	12	Mild DD
9	Fat and protein restriction	1.5	1.7 g/kg/day	30%	Twice daily	4 hrly	UCCS introduced	100 mg/kg/day	1	Normal
10	Fat and protein restriction	0.9	1.8 g/kg/day	30%	Twice daily	4 hrly	Frequent feeds (4 hrly)	100 mg/kg/day	0	Normal

All patients have maltodextrin-based emergency management plans based on 120% estimated energy requirement or, if not tolerated, dextrose intravenously(4.5.1). Rx—treatment. UCCS uncooked cornstarch. DD developmental delay. Hrly—hourly. * Normal neurology and intellect but had occipital infarct with decompensation—previously published [4]. # Intellectual outcomes are based on situation when well. Intellectual function is in relation to peers—e.g., mainstream education without support—note Patient 2 has Usher Syndrome. Further details on all cases in online material.

**Table 3 nutrients-15-00531-t003:** Practice guidelines for cohort of patients identified with HMGCL deficiency.

Age	When Well	When Unwell (All Ages)
**At all ages**	Consider use of L-carnitine and/or S-DL-3OHB Assessment of adequacy of energy, protein, micronutrient, and essential fatty acid intake, with diet modification and supplementation as required	Adequate and regular intake (1–3 hourly) of carbohydrate when unwell to meet energy requirements (~120% estimated energy requirement using appropriate activity factor). 3 h maximum fast duration in infancy and 4 h later in childhood. Consider adjusting dose and/or frequency s of L-carnitine and S-DL-3OHB. When improving clinically, introduce usual diet as tolerated with 4 h maximum fast duration. Reintroduce protein sources within 24–48 h to avoid catabolism.
**Neonatal**	Replace a third of usual breastmilk or infant formula fluid intake with maltodextrin solution at same energy concentration (e.g., 50 mL/kg) resulting in Fat restriction to 33% usual energy intake. (NOTE usual fat consumption of breast/formula fed infants is 40–57% energy intake). Protein restriction of 1–1.3 g/kg Avoid fasting with 3–4 hourly feeding.	
**Infant**	Maintain maltodextrin solution and breastmilk/formula proportions. Encourage regular intake of low-fat (< 3g fat/100 g), carbohydrate-rich foods. (providing ~10% energy intake from fat). Initially avoid high-protein foods (animal products and high-protein alternatives). Gradually increase overnight fasting interval from 6 months of age.	
**From one year of age**	As breastmilk/infant formula intake decreases, consider introducing small serves of low-fat, high-protein foods (e.g., in 5 g protein serves.) Introduce low-protein, low-fat milk alternatives. Introduce concept of low-glycemic-index foods. Consider introduction of nocte 1–2 g/kg uncooked cornstarch (or modified cornstarch product from 2 years of age) to extend overnight fasting time.	
**Older children**	Consider liberalization of fat and protein with guidance on healthy fat and protein choices. (NOTE Australian healthy eating guidelines for fat intake in childhood is ~30% energy intake). Avoid extended fasts. Adequate carbohydrate and protein intake for high-intensity and/or prolonged exercise. Avoid prolonged overnight fast (continue nocte cornstarch if required)	
**Teenagers/adults**	Explain reason for fasting avoidance. Explain unwell management/pregnancy plan. Consider nocturnal cornstarch. Avoid excessive alcohol.	

## Data Availability

Not applicable.

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
