# Peer review of "Treatment of HMG-CoA Lyase Deficiency—Longitudinal Data on Clinical and Nutritional Management of 10 Australian Cases"

_nutrients, 2023, doi:10.3390/nu15030531_

Round 1

Reviewer 1 Report

See attached file

Author Response

management of ten Australian cases” is a clear and well-structured manuscript. However, I have some question and change suggestions.

  1. When you discuss the Practice guidelines for the management of the HMGCL deficiency, you do not say anything about the supplementation with L-carnitine, although it is a recurrent treatment in your cohort. Could you please explain the rationality behind the administration of L-carnitine to these patients, the doses and the clinical or biochemical data, which should be taken into account?

A new section in treatment has been added:

4.5.4 There is no published consensus on optimal carnitine dose and this variation is reflected in the Australian cohort. Similar to other inborn errors of metabolism such as glutaric aciduria or methylmalonic aciduria, 100mg/kg/day of L-carnitne is typically used in childhood, which is decreased, in terms of dose/kg, in older individuals.

  • I completely understand the value of including the first reported patient with HMGCL deficiency. However, you do not have any data about the treatment he is following nowadays or even if he is still alive. Therefore, I suggest either trying to obtain these data or deleting the patient from the study.
  • We were unable to locate the next of kin for the index case for consent purposes but are aware that the diagnostic laboratory received samples until he was aged 35 years old. We have added information relating to this but are unable to provide further clinical information, without consent. The authors feel it is important to include this case as it has helped shape awareness in Australia and indicates that survival is possible, even from the first case. It also indicates that adult specialist clinical services are required to manage this condition as the case is presumed to have died at some point over the age of 35 years.

This has been added to the conclusion:

Lifelong vigilance and specialist care is required for this disorder as there are a number of fatalities that have been reported at different ages.

There are some minor changes, which should be corrected:

  1. Title in table 1 is missing - done
  2. Lines 148 (from Rx) to 151 should be at the end of Table 2 done
  3. Line 55: change 3-hydroxy-3-methylglutyrate to 3-hydroxy-3-methylglutyrate done

Reviewer 2 Report

There are several changes that need to be made, to improve clarity, and since the report may be used to guide treatment of those affected by HMGCoA lyase it's important that the evidence for the recommendations is outlined; i.e. to make it clear to what extent the guidelines are based on the experience/opinion of the authors and how much on independent studies (publications).

I presume the results of the study, Table 1, Table 2 and text in the results and discussion section is for patients that presented to emergency and were resuscitated, presumably because of metabolic decompensation. Section 1.3 specifies “Since 2010 in NSW, patients with HMGCL deficiency have been resuscitated in emergency department”.  Section 2, implies the data for study are from 5 centres in Australia. Are these from NSW, and are the results in Table 1 based on their presentation to emergency? Were they likewise resuscitated as implied in the preceding section (Section 1.3)? Or were they very unwell, critically unwell, metabolically decompensated but not requiring resuscitation-I’m putting an emphasis on the term resuscitation here. I found this very confusing.

Line 101- Change “Patient 4 was also published”, to “studies on patient 4 was also published”. 

Line 107- Be consistent:  change “two months to 39 years” to “2 months to 39 years”.

Line 26-Delete “Newborn screening” as a title. This title is not necessary and is misleading. The paragraph is not about newborn screening ,and it disrupts the flow.

Table 2. “ all patients have sick day plan” should be changed to “ all patients have a sick day plan” or “all patients have six day plans”. This Note should also be a footnote underneath the table, it should not be in the title.

Table 2. Some further information is required to add context to some of the column headings. Currently it’s ambiguous and one is left guessing. 

“Current protein and Current Fat” pertains to what? I presume after treatment of their acute presentation? The diet recommended/prescribed to the patient for ongoing management?

“Intellectual Outcomes”: qualify how intellectual Outcomes was assessed. And specify the time frame: for example is this in context of the treatment they received after presenting to emergency; or later? It isn’t clear.

Second column, Initial Rx. How does “Initial” differ from “Current”. “Initial” is when and “Current” is when?

Section 4.2-should this section not be in the results?

Section 4.5, Line 244. Specify in the text the published evidence used to support the guidelines. In addition, should “our experience” not be changed to “our opinion, based on experience”?

Table 3. Specify, in the table, which of the guidelines is supported by which published guideline and which on the authors opinion.

Table 3. Correct formatting for quantities. For example, change 50ml/Kg to 50 ml/Kg, 1-1.3g/Kg to 1-1.3 g/Kg etc.  

Grammar. Check grammar throughout. E.g. Line 161 implies stable isotope studies were done in this study. Line 203 relating to MRI spectroscopy needs greater context. It adds little in its current state. Line 287, “hrly” is not an accepted abbreviation. Change to “4 hourly”.

Author Response

There are several changes that need to be made, to improve clarity, and since the report may be used to guide treatment of those affected by HMGCoA lyase it's important that the evidence for the recommendations is outlined; i.e. to make it clear to what extent the guidelines are based on the experience/opinion of the authors and how much on independent studies (publications). We have made a number of changes throughout the manuscript which indicate what is reported and what is our experience:

Eg Stable isotope studies were performed reported in HMGCL affected non-identical twins

I presume the results of the study, Table 1, Table 2 and text in the results and discussion section is for patients that presented to emergency and were resuscitated, presumably because of metabolic decompensation. Section 1.3 specifies “Since 2010 in NSW, patients with HMGCL deficiency have been resuscitated in emergency department”.  Section 2, implies the data for study are from 5 centres in Australia. Are these from NSW, and are the results in Table 1 based on their presentation to emergency? Were they likewise resuscitated as implied in the preceding section (Section 1.3)? Or were they very unwell, critically unwell, metabolically decompensated but not requiring resuscitation-I’m putting an emphasis on the term resuscitation here. I found this very confusing.

We appreciate the reviewer’s frustration and have reorganised as follows:

We have moved the reference to ketones in NSW into history section 1.1 as this is published and referenced. We have added a statement that the management is now Australia –wide policy. The issue about resuscitation is in Discussion 4.5.3:

High dose ketones are used both in acute resuscitation and pre-emptively in the sick day protocol to prevent decompensation.

Line 101- Change “Patient 4 was also published”, to “studies on patient 4 was also published”.

Done

Line 107- Be consistent:  change “two months to 39 years” to “2 months to 39 years”.

Done

Line 26-Delete “Newborn screening” as a title. This title is not necessary and is misleading. The paragraph is not about newborn screening ,and it disrupts the flow.

The authors feel that there is an impact of newborn screening and have changed the title to “The impact of Newborn Screening.” We have summarised the impact in the following sentence:

All four patients that were identified by newborn screening have subsequently presented with acute metabolic decompensation, including hypoglycemia and acidosis indicating that prospective diagnosis has allowed pre-emptive treatment.

Table 2. “ all patients have sick day plan” should be changed to “ all patients have a sick day plan” or “all patients have six day plans”. This Note should also be a footnote underneath the table, it should not be in the title.

We have changes this “All patients have maltodextrin based emergency management plans based on 120% estimated energy requirement or if not tolerated dextrose intravenously.”

Table 2. Some further information is required to add context to some of the column headings. Currently it’s ambiguous and one is left guessing.

Made some changes

“Current protein and Current Fat” pertains to what? I presume after treatment of their acute presentation? The diet recommended/prescribed to the patient for ongoing management?

“intake” added

“Intellectual Outcomes”: qualify how intellectual Outcomes was assessed. And specify the time frame: for example is this in context of the treatment they received after presenting to emergency; or later? It isn’t clear.

This has not been standardised across the whole age range and jurisdictional boundaries. The following has been added to qualify the data. 

# Intellectual outcomes are based on situation when well. Intellectual function is in relation to peers – eg mainstream education without support. Note patient 2 has Usher Syndrome. Further details on all cases in online material.

Second column, Initial Rx. How does “Initial” differ from “Current”. “Initial” is when and “Current” is when?

See change

Section 4.2-should this section not be in the results?

the clinical presentation data is presented in results and this discussion highlights atypical presentations expanding upon the detail within 4.2 – this now specifically discusses atypical presentation.

Section 4.5, Line 244. Specify in the text the published evidence used to support the guidelines. In addition, should “our experience” not be changed to “our opinion, based on experience”?

Changed

Table 3. Specify, in the table, which of the guidelines is supported by which published guideline and which on the authors opinion.

There are no published guidelines for the treatment of this condition which is why we felt compelled to write our clinical nutritional experience for this very severe disorder. This is the basis of submission to this edition of this journal. The guideline discusses the Australian experience, for which we have data– it is difficult to glean this detail from other publications indicating a publication gap.

Table 3. Correct formatting for quantities. For example, change 50ml/Kg to 50 ml/Kg, 1-1.3g/Kg to 1-1.3 g/Kg etc. 

Grammar. Check grammar throughout. E.g. Line 161 implies stable isotope studies were done in this study. Changed to reported

 Line 203 relating to MRI spectroscopy needs greater context. It adds little in its current state. –

The sentence has been changed to:

HIVA and 3-HMG have also been observed in patients with MRI spectroscopy, indicating that these proximal metabolites (figure 1) could be involved in pathophysiology.

Line 287, “hrly” is not an accepted abbreviation. Change to “4 hourly”. Done
